# ITS Metabarcoding Reveals the Effects of Oregano Essential Oil on *Fusarium oxysporum* and Other Fungal Species in Soil Samples

**DOI:** 10.3390/plants12010062

**Published:** 2022-12-22

**Authors:** Lefkothea Karapetsi, Emmanouil Pratsinakis, Fotis Xirakias, Maslin Osathanunkul, Ioannis Vagelas, Panagiotis Madesis

**Affiliations:** 1Laboratory of Molecular Biology of Plants, Department of Agriculture Crop Production and Rural Environment, University of Thessaly, Fytokou St., 38446 Nea Ionia, Greece; 2Institute of Applied Biosciences (INAB), Centre for Research and Technology Hellas (CERTH), 6th Km Charilaou-Thermi Road, 57001 Thessaloniki, Greece; 3Laboratory of Agronomy, School of Agriculture, Faculty of Agriculture, Forestry and Natural Environment, Aristotle University of Thessaloniki, 54124 Thessaloniki, Greece; 4Department of Biology, Faculty of Science, Chiang Mai University, Chiang Mai 50200, Thailand; 5Research Center in Bioresources for Agriculture, Industry and Medicine, Chiang Mai University, Chiang Mai 50200, Thailand; 6Laboratory of Plant Pathology, Department of Agriculture Crop Production and Rural Environment, University of Thessaly, Fytokou St., 38446 Nea Ionia, Greece

**Keywords:** oregano essential oil, *F. oxysporum*, ITS metabarcoding, soil fungal interactions

## Abstract

The destructive effects of *Fusarium* wilts are known to affect the production of many crops. The control of *Fusarium oxysporum* and other soilborne pathogens was mainly based on soil fumigation (methyl bromide), which has long been prohibited and, nowadays, is based on a limited number of available fungicides due to legislation restrictions on residue tolerances and environmental impacts. Alternatively, natural and environmentally safe compounds, such as essential oils, are being investigated for their efficacy in the control of soilborne diseases. The great fungicidal ability of the oregano essential oil components (carvacrol and thymol) has been reported to inhibit the germination and the mycelial development of several fungal species, including *F. oxysporum*. The aim of our study was to demonstrate how the metabarcoding approach can provide valuable information about the positive or negative impacts of two different doses of oregano essential oil on *Fusarium oxysporum* and other fungal species which were present in the studied soil samples through the amplification of the ITS1 and ITS2 regions, which were analyzed on a MiSeq platform. A higher dose of oregano essential oil decreased the abundance of *F. oxysporum*, along with other fungal species, but also had negative effects on *Trichoderma evansii* and *Mortierella chlamydospora*, species with possible fungicidal properties. Soil properties, essential oil properties, the fungal composition, and interactions between fungal species should be considered as factors influencing the effectiveness of essential oils as biological control agents for soilborne pathogens.

## 1. Introduction

*Fusarium oxysporum* is among the world’s most damaging plant pathogens, capable of infecting nearly all economically important crops and causing billions of dollars’ worth of losses in global agriculture every year [1]. As a soilborne fungus, it is responsible for vascular wilt, rot, and damping-off diseases in a broad range of plants. *Fusarium oxysporum* (wilt) can survive in the soil between crop cycles in infected plant debris. The fungus can survive either as mycelium or as any of its three different spore types. The roots can be infected directly through the root tips or wounds. Once inside the plant, the mycelium grows through the root cortex until it reaches the xylem and, later, throughout the vascular tissue of the whole plant. This condition progressively limits water and nutrient uptake, so that the leaves wilt and the plant eventually dies [1].

Conventional control strategies for soilborne pathogens rely mostly on chemical fungicide applications for the majority of crops and less on resistant cultivars [2]. Disease management is associated with the high cost of the fungicides, development of pest resistance, growing environmental concerns about the use of fungicides, and difficulty in identifying resistant cultivars. All these issues have resulted in an increase in the number of studies investigating the antimicrobial properties of plant extracts and main essential oils [3,4]. Many antifungal compounds from plants have been reported to control pathogens [5]. There is a significant number of studies reporting on the antimicrobial and antifungal properties of plant extracts and essential oils for the protection of food and grain storage against foliar, soilborne pathogens, and nematodes [4,6,7,8]. The effectiveness of plant essential oils has been assessed mainly in vitro, and their efficiency in maintaining soil health has only gained interest during the last few years [3,9,10,11,12].

Essential oils are by-products of plant metabolism and are commonly referred to as secondary plant metabolites, with their most important properties being related to the plants’ adaptation to abiotic stresses, intra-plant and inter-plant signaling, and direct and in-direct defense against herbivores and pathogens [13]. Other uses of essential oils have also been reported, including their use as food additives, flavorings, and cosmetic ingredients [4]. The three biosynthetic groups of essential oil components are terpenes, phenylpropanoids, and isothiocyanates [14]. Their natural origins provide improved safety for both human health and the environment [11].

The antifungal properties of essential oils have also been investigated and found to be effective for many fungal species, such as *Botrytis cinerea*, *Rhizoctonia solani*, *Fusarium moniliforme*, *Sclerotinia sclerotiorum*, *F. oxysporum*, *Aspergillus niger*, *A. flavus*, *Penicillium digitatum*, *Pythium ultimum*, and *Phytopthora* sp. [11,15]. Carvacrol, the main compound of the oregano essential oil, has been found to be one of the most effective metabolites against plant pathogens. Carvacrol’s mechanism of action is correlated with its ability to disrupt the integrity of cell walls and membranes and inhibit germination and mycelial development [16,17,18]. Several studies have also noted the absence of phytotoxicity in oregano essential oil used for the control of insects and fungi [18].

Yet, there is a lack of studies regarding the effects of essential oils on the abundance of other soil micro-organisms, not only in vitro but also in soil, where the organic matter and other soil properties may have impacts [19].

The development of next-generation sequencing technologies has greatly improved our ability to identify any species, including fungi. Fungal communities in soil can have positive or negative impacts on both crop plant health and the environment [20]. Studying and understanding the complexity of fungal communities require the accurate identification of the present fungal species. This task is characterized by many challenges, especially due to the morphological variability of many fungal species [21]. The sequence-based characterization of fungal community species, or metabarcoding, has been the standard technique for the last decade [22,23]. For fungi, the internal transcribed spacer (ITS) region in the ribosomal RNA (rRNA) operon has been accepted as the formal and universal fungal barcode for species-level identification [24] and is widely used to sequence fungal communities [25], in combination with large taxonomic reference databases, such as UNITE [26]. Soil represents a complex environment, and DNA metabarcoding is widely recognized to provide us with a more unbiased, sensitive, and broad method of detection of all the present fungal species compared to traditional isolation-based strategies.

During our research, we investigated how metabarcoding can reveal the effect of oregano essential oil on *Fusarium oxysporum* in inoculated soil samples, along with possible effects on the general fungal composition of inoculated and non-inoculated soil samples, through the next-generation sequencing of the amplicons ITS1 and ITS2.

## 2. Results

### 2.1. GC Analysis

The main constituents observed by gas chromatography (GC) in the Origanum vulgare essential oil were Carvacrol (51.5%), p-Cymene (19.9%), γ-Terpinene (8%), Thymol (3.29%), β-Caryophyllene (2.45%), β-Myrcene (2.24%), α-Terpinene (1.5%), Sabinene (1.24%), α-Pinene (1.2%), β-bisabolene (1.2%), d-limonene (0.97%), borneol (0.84%), Carvacrol methyl ether (0.65%), α-Terpineol (0.58%), β-Pinene (0.35%), and α-Caryophyllene (0.29%).

### 2.2. Sequencing

ITS amplicon sequencing on the MiSeq platform produced a total of 766,450 sequences for all the samples of a high quality. According to Table 1, a higher number of reads was obtained for the A20 samples (non-inoculated, treated with 20% oregano essential oil), followed by the A0 samples (non-inoculated, not treated with oregano essential oil). Lower numbers of reads were obtained for the M5 samples (inoculated with *F. oxysporum*, treated with 5% oregano essential oil) followed by M20 (inoculated with *F. oxysporum*, treated with 20% oregano essential oil). After trimming and filtering, the number of reads was not significantly altered, with a total of 763,149 reads, as the number of sequences containing ambiguous bases was low among all the samples, with an average of 127,192 reads/sample. These reads were classified into 1619 unique OTUs that were assigned as fungi.

### 2.3. Relative Abundance of Soil Fungal Community per Treatment

A percentage of almost 75–80% of the total sequences identified per sample was assigned to the kingdom of fungi. According to the phylum abundance in all the A samples, Ascomycota was the most abundant, with a percentage of almost 80%, and lower percentages were observed in the case of Mortierellomycota (11.14%) and Basidiomycota (8.98%). In all the M samples, Ascomycota was the most abundant, with a lower percentage, compared to the A samples, with almost 50%, followed by Mucoromycota (36.89%), Mortierellomycota (11.8%), and Basidiomycota (1.61%) (Figure 1).

The relative abundance analysis of the class revealed the presence of fungi belonging mostly to Eurotiomycetes (38.84%), Sordariomycetes (29.12%), Mortierellomycetes (11.14%), and Orbiliomycetes (10.5%), with the least belonging to Agaricomycetes (6.8%) and Pucciniomycetes (2.1%), in all the non-inoculated samples. In all the inoculated samples, the most abundant classes were those of Sordariomycetes (41.56%) and Mucoromycetes (36.82%), followed by Mortierellomycetes (11.8%), Eurotiomycetes (6.22%), and Tremellomycetes (1.52%) (Figure 2).

In the A samples, the relative abundance analysis revealed the presence of orders belonging mostly to Eurotiales (27.73%), Pezizomycotina (15.6%), Mucorales (11.9%), and Conioscyphales (10.45%) and less belonging to Pezizales, Agaricales, Hypocreales, Microascales, and Pucciniales. For the M samples, the most abundant orders were Pezizomycotina (34.68%) and Microascales (32.51%), followed by Hypocreales, Eurotiales, Mortierellales, Microstamatales, and Pucciniales (Figure 3).

### 2.4. Relative Abundance (Family and Genus) per Sample for Each Treatment

According to relative abundance analysis, in the A0 sample, the most abundant families were Ajellomycetaceae (18.25%), Mortierellaceae (14.56%), Orbiliaceae (10.98%), Conioscyphaceae (10.34%), Microascaceae (10.12%), and Cephalothecaceae, with Thermoascaceae, Aspergillaceae, and Pucciniaceae in lower percentages (Figure 4A). In the M0 sample, where the compositions of the families differed significantly compared to the A0 sample, the most abundant families were Rhizopodaceae (30.62%), Nectriaceae (23.01%), Mortierellaceae (17.89%), Aspergillaceae (7.34%), and Hypocreaceae (7.16%), with Microascaceae at a lower percentage (Figure 4B).

The addition of oregano essential oil affected the percentages of the families present in the A5 and M5 samples. In sample A5, the most abundant families were Ajellomycetaceae (27.33%), Orbiliaceae (16.13%), Microascaceae (12.23%), and Cephalothecaceae (7.07%), and in lower percentages, Thermoascaceae, Mortierellaceae, and Conioscyphaceae (Figure 5A). In sample M5, the most abundant families were the same as those in sample M0, but Rhizopodaceae showed an increase of 6% and Nectriaceae showed an increase of 12%, while on the other hand, Mortierellaceae showed a decrease of almost 11%, while Aspergillaceae showed a decrease of 5% and Hypocreaceae showed a decrease of only 1% (Figure 5B).

The addition of oregano essential oil at a higher dose also affected the fungal composition. In sample A20, Ajellomycetaceae was further decreased to 20% compared to sample A5, while the percentage of Psathyrellaceae increased to 14.38% from 2.69% in sample A5, the percentage of Mortierellaceae increased from 5.89% to 12.11% in sample A20, Cephalothecaceae was not significantly affected, and the percentages of Microascaceae and Orbiliaceae were decreased to 7.9% and 5.31%, respectively. Aspergillaceae and Pucciniaceae were not affected by the higher dose of oregano oil (Figure 6A). Compared to sample A20, the presence of *Fusarium oxysporum* and the higher dose of oregano essential oil in sample M20 did not significantly alter the presence of Rhizopodaceae (39.07%), Mortierellaceae (9.42%) or Hypocreaceae (5.35%), while the percentage of Nectriaceae was decreased by almost 15%. The percentages of Mucoraceae and Aspergillaceae were increased compared to sample M5 (Figure 6B).

The differences in the fungal composition on the genera level between the non-inoculated and inoculated samples and how the addition of oregano essential oil affected the fungal abundance are demonstrated in Appendix A.

### 2.5. Species Diversity and Effect of Oregano Essential Oil

The different treatments demonstrated differences in the species richness and evenness indices, such as the number of observed OTUs, Chao1 (an abundance-based estimator), and the diversity indices, including Shannon and Simpson. Generally, higher observed OTU, ACE, and Chao1 indices demonstrated a higher species richness in the case of the non-inoculated treatment, while an increased Shannon index indicated greater microbiome diversity. Specifically, the ACE, Chao1, and observed OTU indices indicated higher species richness in sample A20 compared to the lower species richness in samples A0 and A5, respectively. According to observed OTU, ACE, and Chao1 indices, the samples that received the inoculated treatment were characterized by lower species richness, with the least diverse sample being M5, followed by M20 and M0. The Shannon index showed higher species richness in A0, followed by A20 and A5, in the case of the non-inoculated treatment, while in the case of the inoculated treatment, M0 had a higher species richness, followed by M5 and M20. The Simpson and invSimpson (inverse Simpson) indices revealed a high species evenness in sample A0, followed by A20 and A5. A lower species evenness was revealed among all the samples subjected to the inoculated treatment, with M20 characterized by the lowest value (Figure 7).

To evaluate the structural similarities in the fungal communities between the two treatments, sample clustering based on the taxonomic abundance profiles was used. Furthermore, to calculate a Bray–Curtis similarity matrix, non-metric multi-dimensional Scaling (NMDS) was used (Figure 8). A long-distance spatial separation between the non-inoculated samples and inoculated samples was observed. The non-inoculated samples (A0-A5-A20) were not grouped together with the inoculated samples (M0-M5-M20), as their fungal composition was highly altered by the inoculation with *F. oxysporum* and the addition of the oregano essential oil.

In all the non-inoculated samples, the main species identified were *Emmonsiellopsis coralliformis*, *Mortierella chlamydospora*, *Arthrobotrys amerospora*, *Chrysosporium pseudomerdarium*, *Conioscypha minutispora*, *Phialemonium globosum*, *Aspergillus chlamydosporus*, and *Puccinia amari* (Figure 9). In sample A5, the addition of 5% oregano essential oil mostly favored the growth of *Emmonsiellopsis coralliformis* (27.33% in contrast to 18.25% in A0) and was less favorable to *Arthrobotrys amerospora* (16.13% in contrast to 10.98% in A0), *Pseudallescheria boydii* (8.3% in contrast to 7.35% in A0), *Byssochlamys zollerniae* (6.07% in contrast to 4.33% in A0), and *Puccinia amari* (2.14% in contrast to 1.56% in A0). On the other hand, 5% oregano essential oil had negative effects on *Mortierella chlamydospora* (5.85% from 14.48% in A0), *Chrysosporium pseudomerdarium* (7.36% from 10.93% in A0), *Conioscypha minutispora* (5.73% from 10.34% in A0), *Phialemonium globosum* (6.29% from 7.6% in A0), and *Aspergillus chlamydosporus* (1.92% from 3.09% in A0). The higher dose of oregano essential oil (20%) mostly favored the growth of *Lacrymaria subcinnamomea* (14.38%, while in samples A0 and A5, it was less than 1%), and had lesser effects on *Mortierella chlamydospora* (12.01% from 5.85% in A5), *Chrysosporium pseudomerdarium* (9.45% from 7.36% in A5), and *Conioscypha minutispora* (7.53% from 5.73% in A5), while it had a negative effect on *Arthrobotrys amerospora* (5.31% from 16.13% in A5) and *Byssochlamys zollerniae* (3.4% from 6.07% in A5). *Aspergillus chlamydosporus* and *Puccinia amari* were not significantly affected by the higher dose of oregano essential oil in the case of the non-inoculated treatment. The *Trichoderma evansii* percentages were below 1% for all the non-inoculated samples.

In all the inoculated samples, the main species identified were *Rhizopus arrhizus*, *Mortierella chlamydospora*, *Neocosmospora rubicola*, *Fusarium oxysporum*, and *Trichoderma evansii* (Figure 10). In both samples M5 and M20, the addition of oregano essential oil mostly favored the growth of *Rhizopus arrhizus* (from 30.62% in M0 to 36.15% in M5 and 39.07% in M20). Another species that was favored by the addition of essential oil was *Myxocephala albida* (8.77% in M20 from 2.29% in M5 and 1.69% in M0) and *Chrysosporium pseudomerdarium* (below 1% in M0, 2.92% in M5, and 2.01% in M20). In the case of *Fusarium oxysporum*, an increase in the percentage was observed in sample M5 (27.06%) with the addition of oregano essential oil, in comparison to sample M0 (8.57%), and a significant decrease to 13.51% was observed in M20, to which a higher dose was applied. Additionally, the higher dose of oregano essential oil had negative effects on *Trichoderma evansii* (5.35% in M20 in contrast to 6.81% in M5 and 7.1% in M0), *Mortierella chlamydospora* (8.32% in M20 and 6.19% in M5 in contrast to 16.24% in M0), and *Neocosmospora rubicola* (2.97% in M20 and 5.7% in M5 in contrast to 13.08% in M0). *Aspergillus inflatus* was also negatively affected (1.39% in M0 and below 1% in M5 and M20). *Puccinia amari* was almost eradicated in samples M5 and M20, to which oregano essential oil was applied (below 1% in M0 and almost 0% in M5 and M20).

## 3. Discussion

Soil-associated microbes are known to play important roles in aspects of soil and plant health such as nutrient uptake and the mitigation of abiotic stresses, such as drought, salt, and adverse temperatures, in addition to contributing to plant immunity and altering the plant gene expression, among other effects [27,28]. In agricultural systems, plants, beneficial bacteria, plant pathogens, and parasites compete for soil resources. Thus, pathogens can negatively affect the quality and yield of crops [29]. Farmers mostly apply chemical pesticides, including fungicides, due to their simple and clear use, along with their effectiveness against pathogens [30]. However, their harmful effects on the environment and human health, along with the development of resistance, have drawn attention and led to the investigation of pathogen management strategies based on natural compounds, such as essential oils [31,32,33].

The antimicrobial and antifungal effects of plant extracts and essential oils on plant pathogens have been known for many years [34]. Nevertheless, during the last 20 years, many secondary compounds from plants have been tested in agriculture for the purpose of pest management [4,35,36].

Oregano essential oil have been shown to inhibit the growth of soilborne pathogens and plant parasitic nematodes. Additionally, during a field experiment based on corn and cotton plants, oregano essential oil was found to affect soilborne pathogens and inhibit their growth [18]. However, a limited number of studies have investigated the effects of oregano essential oil on the fungal communities in soil samples with or without the presence of soilborne pathogens, such as *Fusarium oxysporum*.

The high-throughput sequencing methodology used in the current study enabled the detection of a range of fungal taxa in the soil. Moreover, it revealed how these taxa are affected both by the presence of a soilborne pathogen such as *F. oxysporum* and the application of oregano essential oil. The composition and the abundance of the fungal communities in the samples that were not inoculated with *Fusarium oxysporum* and samples inoculated with *Fusarium oxysporum* were significantly different, even without the addition of the oregano essential oil. The non-inoculated samples were characterized by higher values of the *alpha* and *beta* diversity indices compared to the inoculated samples. The higher values of the observed OTU, ACE, Chao1, Shannon, Simpson, and invSimpson indices indicated higher species richness and evenness in samples A0, A5, and A20, respectively. The inoculation with *F. oxysporum* was found to affect the compositions of other fungal species that were already present in the soil samples, as shown by the comparisons between the non-inoculated and inoculated samples, especially in the case of samples A0 and M0, to which no oregano essential oil was added. In particular, fungi belonging to Ascomycota accounted for almost an 80% of total fungi in the case of the non-inoculated treatment, while in the inoculated samples, the percentage dropped to almost 50%, and the number of fungi belonging to Mucoromycota increased to almost 37% (Figure 1). Moreover, the composition of fungi according to the class and order was altered between the non-inoculated and inoculated samples, whereas the percentage of Sordariomycetes was decreased and the percentage of Mucoromycetes was increased in the inoculated samples (Figure 2 and Figure 3). According to the composition analysis, in the inoculated samples, Rhizopodaceae, Nectriaceae, and Mortierellaceae were the most abundant families, in contrast to the non-inoculated samples, in which Rhizopodaceae and Nectriaceae were present in very low percentages (below 1%). This fact can be partially explained by the fact that *Fusarium oxysporum* belongs to the Nectriaceae family, and *Fusarium* could only be present in the inoculated samples (M samples). This was also confirmed by the genus and the species composition analysis, which revealed an increase in *Neocosmospora rubicola*, which also belongs to Nectriaceae (Appendix A, Figure 10). Furthermore, the species abundance analysis revealed the presence of *Emmonsiellopsis coralliformis*, *Mortierella chlamydospora*, *Arthrobotrys amerospora*, *Chrysosporium pseudomerdarium*, *Conioscypha minutispora*, *Phialemonium globosum*, *Aspergillus chlamydosporus*, and *Puccinia amari* in the non-inoculated samples, while in the inoculated samples, *Rhizopus arrhizus*, *Mortierella chlamydospora*, *Neocosmospora rubicola*, *Fusarium oxysporum*, and *Trichoderma evansii* were identified as the most abundant species. Thus, the inoculation of the soil samples with *F. oxysporum* appeared to increase the abundance of the *Rhizopus arrhizus*, *Neocosmospora rubicola*, and *Trichoderma evansii* species. The *Trichoderma* species are characterized as a highly opportunistic species and, most frequently, as conidial fungi that can be isolated from a diverse range of natural and artificial substrates [37]. Their ability to antagonize plant-pathogenic fungi and adapt to various environmental conditions has been confirmed by genomic analysis [38]. Limited information is available for *Neocosmospora rubicola* concerning its interaction with *F. oxysporum*, apart from the fact that it was included in the genus *Fusarium* until recently [39]. Several studies have reported the wide-ranging presence of *Rhizopus arrhizus* in many organic substrates and soil samples, as well as its potential fungicidal properties against plant pathogens such as *Fusarium oxysporum* [40,41]. Moreover, the microbial community is mostly influenced by the synergies and/or competition between the numerous microbial members that perform functions for the plant’s health as a whole [42,43]. Thus, the inoculation of soil *with F. oxysporum*, as we observed here, altered the fungi composition and the percentages of the different fungi in the soil. This should be expected, as the addition of a new species influences the microbiome community of the soil sample, introducing new relationships between the members of the community. New studies have shown that microbiomes in certain areas are very well defined, shaping complex interconnected microbial networks, in which each microbial species interacts with the others directly or indirectly through processes such as competition, facilitation, and inhibition [44,45,46,47,48,49] in a positive or negative manner in the established microbiome, and new connections can influence the composition of the given microbial community. Furthermore, here, the addition of oregano essential oil was shown to affect the overall fungal composition of the soil samples, along with the effect on *F. oxysporum*. The percentage of *F. oxysporum* was almost 25% in the soil sample to which 5% oregano essential oil was added. With the increase in the concentration of oregano oil to 20%, a decline to 13% was detected (Figure 10). The antifungal properties of oregano essential oil have already been reported in regard to *Fusarium spp*. and other plant pathogenic fungi, such as *Alternaria alternata*, *Botrytis cinerea*, *Macrophomina phaseolina*, *Penicillium* spp., *Phytophthora capsici*, *Rhizoctonia solani*, and *Sclerotinia sclerotiorum* [50]. In the non-inoculated samples, the application of oregano essential oil favored the growth of *Emmonsiellopsis coralliformis*, *Arthrobotrys amerospora*, *Pseudallescheria boydii*, *Byssochlamys zollerniae*, and *Puccinia amari*, while it had a negative effect on *Mortierella chlamydospora*, *Chrysosporium pseudomerdarium*, *Conioscypha minutispora*, *Phialemonium globosum*, and *Aspergillus chlamydosporus*. A higher dose of oregano essential oil (20%) mostly favored the growth of *Lacrymaria subcinnamomea*, while it had a negative effect on *Arthrobotrys amerospora* and *Byssochlamys zollerniae*. The higher dose of oregano essential oil was shown to have a negative effect on *Trichoderma evansii* and *Mortierella chlamydospora*. These results should be taken into serious consideration when essential oils are applied for the control of soilborne pathogens, as both species are characterized by their fungicidal potential and their use as biocontrol agents against *F. oxysporum* [37,51]. Although antifungal activity against *Aspergillus* and *Puccinia* species has been reported before in the case of essential oils, in our study, the higher dose of oregano essential oil had no effect on the percentages of *Aspergillus chlamydosporus* and *Puccinia amari* in the non-inoculated soil samples [52]. In contrast, *Aspergillus inflatus* was negatively affected and *Puccinia amari* was almost eliminated in the inoculated samples M5 and M20, to which oregano essential oil was applied.

Even though antifungal properties of essential oils have been recorded, the inhibition of fungal growth is extensively affected by the dose, which mainly varies based on the type and concentration of the compounds of the essential oils [53]. In addition, the physical and chemical properties of the soil, along with the fungal composition of the soil used for testing, have also been reported to affect the antifungal action of oregano essential oil against fungal pathogens and the fungicidal activity of other fungal species [2,54].

## 4. Materials and Methods

Inoculum preparation: A *Fusarium oxysporum* isolate (BFI 2550) obtained from the Benaki Phytopathological Institute (BFI, Athens, Greece) was used in this study. The inoculum was prepared from 3-week-old potato dextrose agar (PDA) cultures of *F. oxysporum*. The PDA plates were incubated at 25 °C for seven days, and after the given incubation period, the plates were flooded with sterile water, and *F. oxysporum* conidia were scraped from the culture with an inoculation loop. The fungus conidial suspension was then filtered through four layers of cheesecloth, quantified with the aid of a hemocytometer, and diluted with sterile water to estimate a concentration of 4 × 10^6^ conidia/mL.

*Origanum vulgare*—Plant material, essential oil isolation, and GC analysis: Aerial parts of *Origanum vulgare* were collected during June–July 2021 from different areas of Greece. The essential oil was obtained by steam distillation using a modified Clevenger apparatus for 3 h. The chemical composition of the essential oil was determined using an Agilent Technologies 7890A gas chromatographer (GC) coupled with an Agilent 5957C VL MS Detector with a triple-axis detector system, mass spectrometer (MS), and flame ionization detector (FID), as described in Evergetis et al. 2016 [55].

Inoculation of the soil samples—treatment with oregano oil: For the infection of the soil, 10 mL of conidial suspension was mixed with 3 kg of soil medium to achieve a concentration of 300:1 *v*/*v* (3 kg of soil medium: 10 mL of fungal conidial preparation estimated at 4 × 10^6^ conidia/mL). The soil medium was a multi-purpose compost (Lidle Grandiol compost). For this experiment, a soil sample was inoculated with the *Fusarium* fungal conidial, while another soil sample was not inoculated, and both were left in sterile plastic bags. Two days later, oregano essential oil, which was encapsulated in β-cyclodextrin at concentrations of 0, 5, and 20%, was added to, and homogenized in, both soil samples. The ratio of soil to oregano essential oil medium was 300:1 *v*/*v* (3 kg of soil: 10 g oregano essential oil encapsulated in β-cyclodextrin preparation at 0, 5, and 20%). The soil water content was maintained as constant by weighing the soil Petri dishes every two days, and the amount of water that was lost was added using spray bottles in a laminar flow hood.

Samples for DNA extraction and amplicon sequencing were collected on the 21st day after the addition of the oregano essential oil at three different concentrations (0, 5, and 20%) in the case of the non-inoculated soil and soil inoculated with *Fusarium oxysporum*. Each soil sample was collected in triplicate.

DNA Extraction: Total DNA from the soil samples was extracted using the NucleoSpin Soil Isolation Kit (Macherey-Nagel, Dueren, Germany) according to the manufacturer’s instructions. The triplicates of DNA from each sample were pooled together with three sample pools of the non-inoculated samples (A0-A5-A20) and three sample pools of the samples inoculated with *Fusarium oxysporum* (M0-M5-M20). The DNA quality and concentration were measured using a Quawell UV–Vis spectrophotometer (Q5000). The extracted DNA was stored at −20 °C until the ITS amplicon libraries were prepared.

Amplicon library preparation: Illumina’s 16S Metagenomics Protocol (part # 15044223 Rev. B) was applied with modifications to amplify the ITS1 and ITS2 regions of the 18S rRNA gene. Two forward and reverse primer pairs containing the Illumina overhang adapters were used for the amplification of the ITS1 and ITS2 regions, respectively. The forward and reverse primer sequences are presented in Table 2. Following the 16S Metagenomics Protocol, 2×Kapa HiFi HotStart ReadyMix was used for the PCR amplification, and the PCR products were then purified with Beckman Coulter Agencourt AMPure XP Beads. The attachment of the indexes and Illumina adapters was performed using the Nextera XT Index Kit, followed by a second cleaning using Beckman Coulter Agencourt AMPure XP Beads. The quantification of the libraries was performed using a Qubit 3.0 fluorometer (Life Technology Ltd., Paisley, UK). In addition, 2 µL of the final libraries were run on a 5200 Fragment Analyzer using the method DNF-473-33-SS NGS Fragment 1–6000 bp to check the quality and the size of the libraries. After the library quantification, normalization, and pooling, the concentrations were adjusted to 12.5 pM and prepared for loading onto the Illumina MiSeq according to the Illumina Metagenomics Protocol. The library pool was denatured and loaded at 6 pM onto Illumina MiSeq and sequenced paired ends (2 × 300) were obtained using the MiSeq Reagent Kitv3 (Illumina, Inc., San Diego, CA, USA).

Bioinformatics and Statistical Analysis: The DADA2 ITS Pipeline Workflow [56] in Galaxy [57,58] was applied for the raw-sequence data processing. Starting with a set of Illumina-sequenced paired-end FASTQ files that had been “demultiplexed” by sample, we obtained an amplicon sequence variant (ASV) table, which is a higher-resolution table than the traditional OTU table. The ASV table demonstrates the number of times that each amplicon sequence variant was observed in every sample that was analyzed. The UNITE database was also implemented for the taxonomy assignment of the output ITS sequences [26].

In particular, the forward and reverse reads were combined into a single FASTA file for each sample, and then the paired-end reads were organized into a paired collection. In contrast to the 16S rRNA gene, the ITS region is highly variable, ranging from 200 to 600 bp in length (Figure 11). The data optimization for the further analysis comprised different steps of trimming and filtering. A key addition to the ITS Workflow is the removal of primer sequences from the forward and reverse reads. For this reason, the Cutadapt tool was used to remove the primers’ sequences from the ITS amplicon sequencing data. The quality control and visualization of the quality profiles of both the forward and reverse readings were performed. Data trimming was performed to remove sequences with ambiguous bases (DADA2 requires sequences with no Ns). As a plausibility verification, the estimated error rates were visualized. The final step of the filtering was the removal of duplicate and chimeric sequences, along with the merging of paired reads. An ASV table was constructed, and the sequence reads were compared to the reference alignment (sh_general_release_dynamic_s_all_10.05.2021.fasta) from the UNITE database using a naïve Bayesian classifier method ([59] kmer size 8 and 100 bootstrap replications). All the relative abundance analyses and the *alpha* and *beta* diversity analyses were based on the ASV spreadsheet file, and visualizations were conducted using the Phyloseq R package (Version: 1.22.3) and PowerBI software (Version: 2.109. 1021.0) [60].

The variation in the fungal communities within and between samples was characterized using *alpha* and *beta* diversity, respectively. The *alpha* diversity assessment was based on the rarefaction curves, measuring the Shannon, Chao1, abundance-based coverage estimator (ACE), and Simpson diversity indices, while the *beta* diversity (Bray–Curtis) was assessed by non-metric multidimensional scaling (NMDS), where the samples were classified and clustered according to their fungal structures.

## 5. Conclusions

The purpose of this study was to evaluate whether the formulated extract of oregano is efficient against *F. oxysporum* when applied to soil and, furthermore, to investigate the effects on the composition of the fungal community already present in the soil used for testing through the metabarcoding approach. The oregano essential oil applied at the higher dose (20%) suppressed the growth of *F. oxysporum* in comparison to the lower dose (5%). However, the decrease was not sufficient compared to the soil sample to which no essential oil was applied. Furthermore, *F. oxysporum* significantly altered the compositions of other fungal species, as demonstrated in the comparison between the non-inoculated and inoculated soil samples (A0 and M0), of which neither were treated with oregano essential oil. Finally, the effects of essential oils on other fungal species with possible antifungal properties, such as those observed here, should be taken into consideration when applying essential oils to soil as antifungal agents. The question of how fungal species respond to the invasion of a pathogen and the interactions that are established between fungal communities remains an open direction for further research.

## Figures and Tables

**Figure 1 plants-12-00062-f001:**
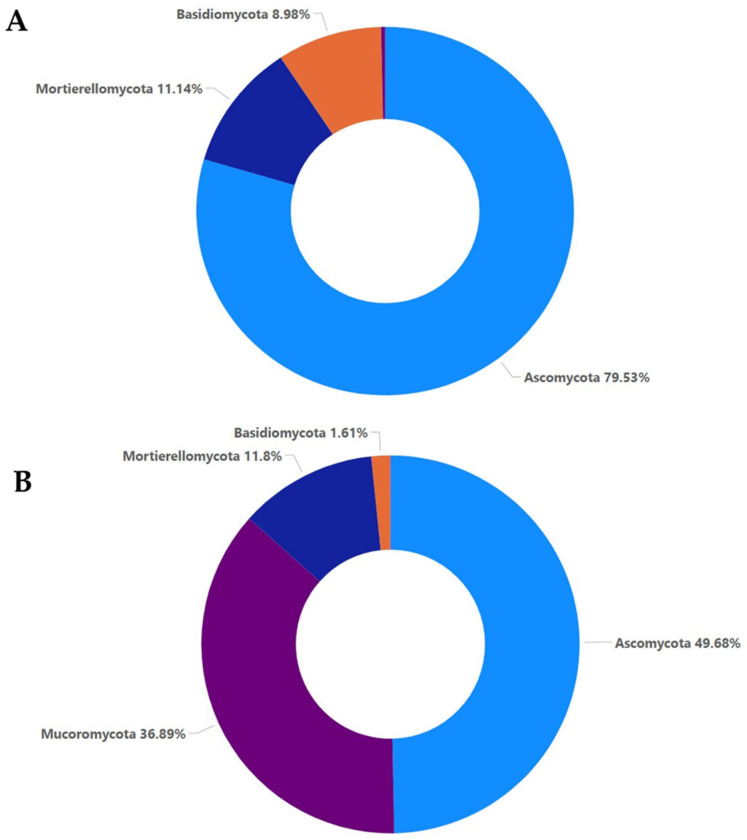
Relative abundance with percentages above 1% for the phylum taxa, as shown for the non-inoculated samples (**A**) and inoculated samples (**B**).

**Figure 2 plants-12-00062-f002:**
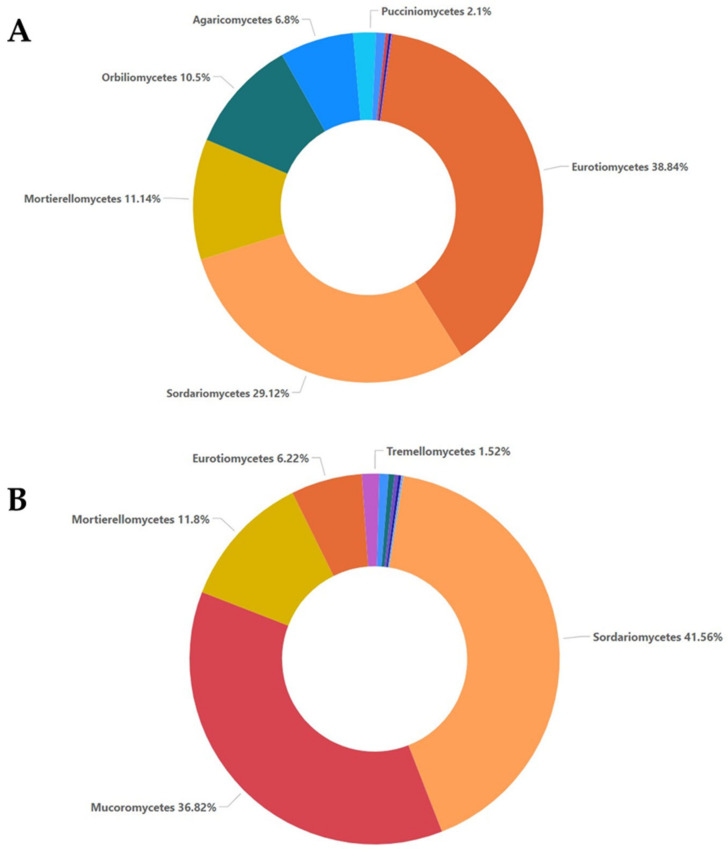
Relative abundance with percentages above 1% for the class taxa, as shown for the non-inoculated samples (**A**) and inoculated samples (**B**).

**Figure 3 plants-12-00062-f003:**
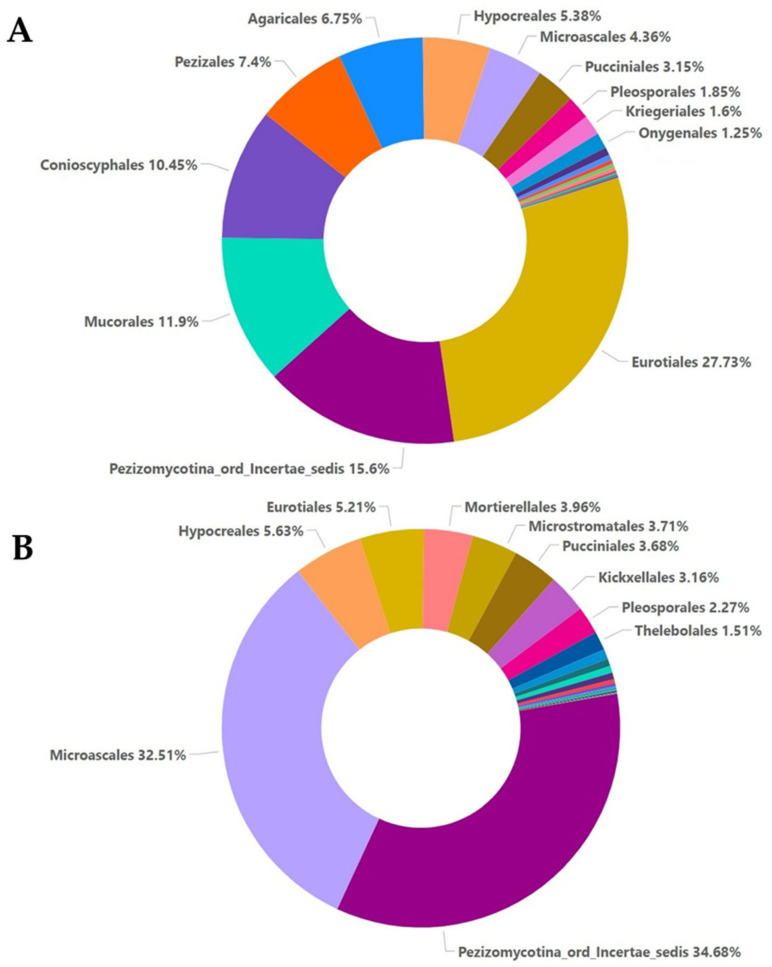
Relative abundance with percentages above 1% for the order taxa, as shown for the non-inoculated samples (**A**) and inoculated samples (**B**).

**Figure 4 plants-12-00062-f004:**
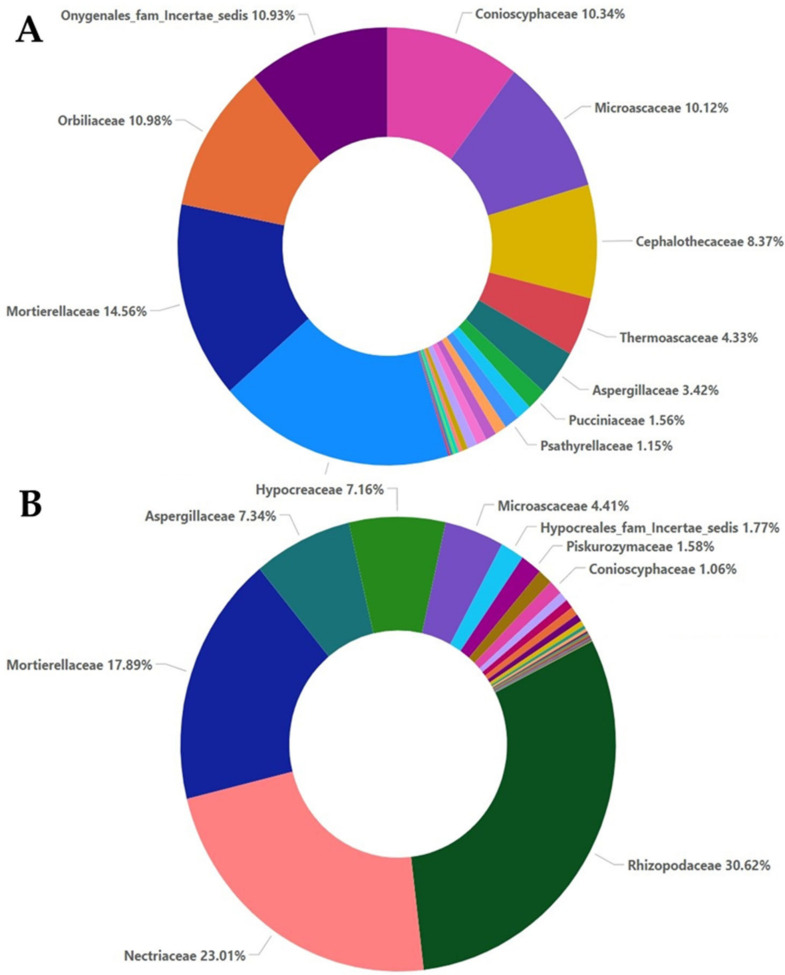
Relative abundance with percentages for the family taxa in the non-inoculated samples (**A**) and inoculated samples (**B**).

**Figure 5 plants-12-00062-f005:**
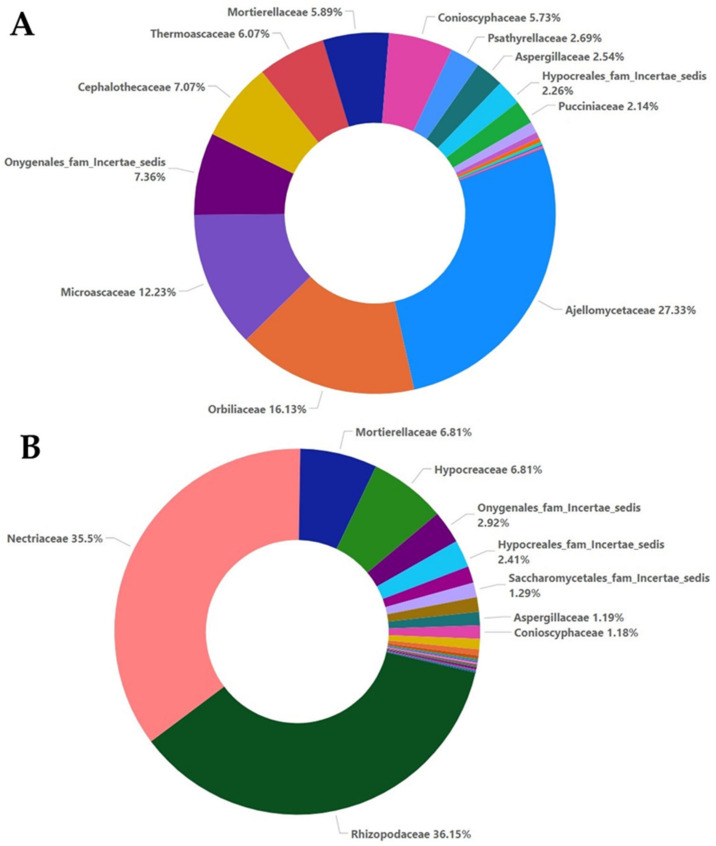
Relative abundance with percentages for the family taxa at a 5% concentration of essential oil for the non-inoculated samples (**A**) and inoculated samples (**B**).

**Figure 6 plants-12-00062-f006:**
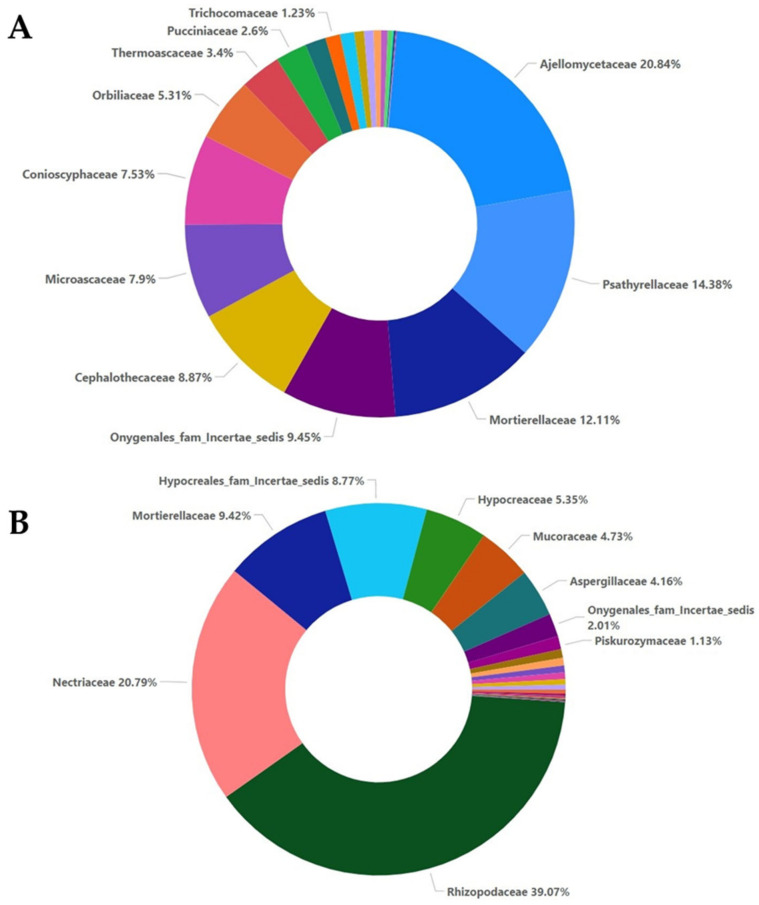
Relative abundance with percentages for the family taxa at a 20% concentration of essential oil for the non-inoculated samples (**A**) and inoculated samples (**B**).

**Figure 7 plants-12-00062-f007:**
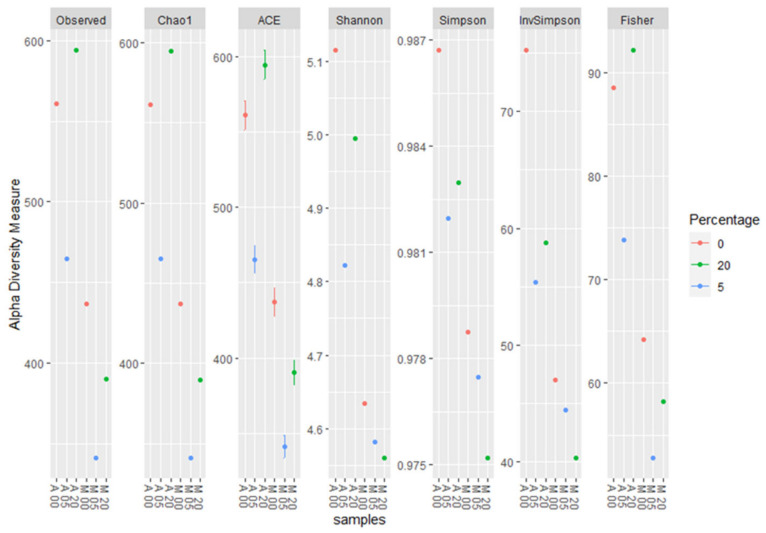
Alpha diversity of the soil fungal community according to the observed OTU, Chao1, Shannon, Simpson, and Fischer indices.

**Figure 8 plants-12-00062-f008:**
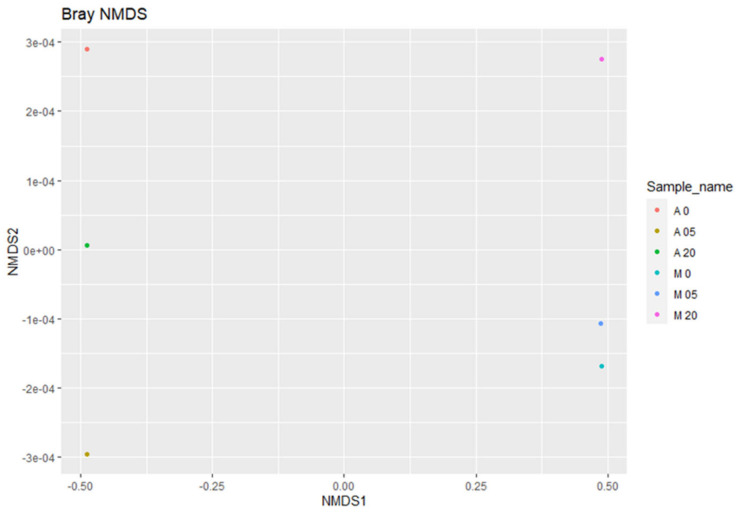
The Bray–Curtis NMDS plot showing the beta diversity of the fungal communities that were identified per sample.

**Figure 9 plants-12-00062-f009:**
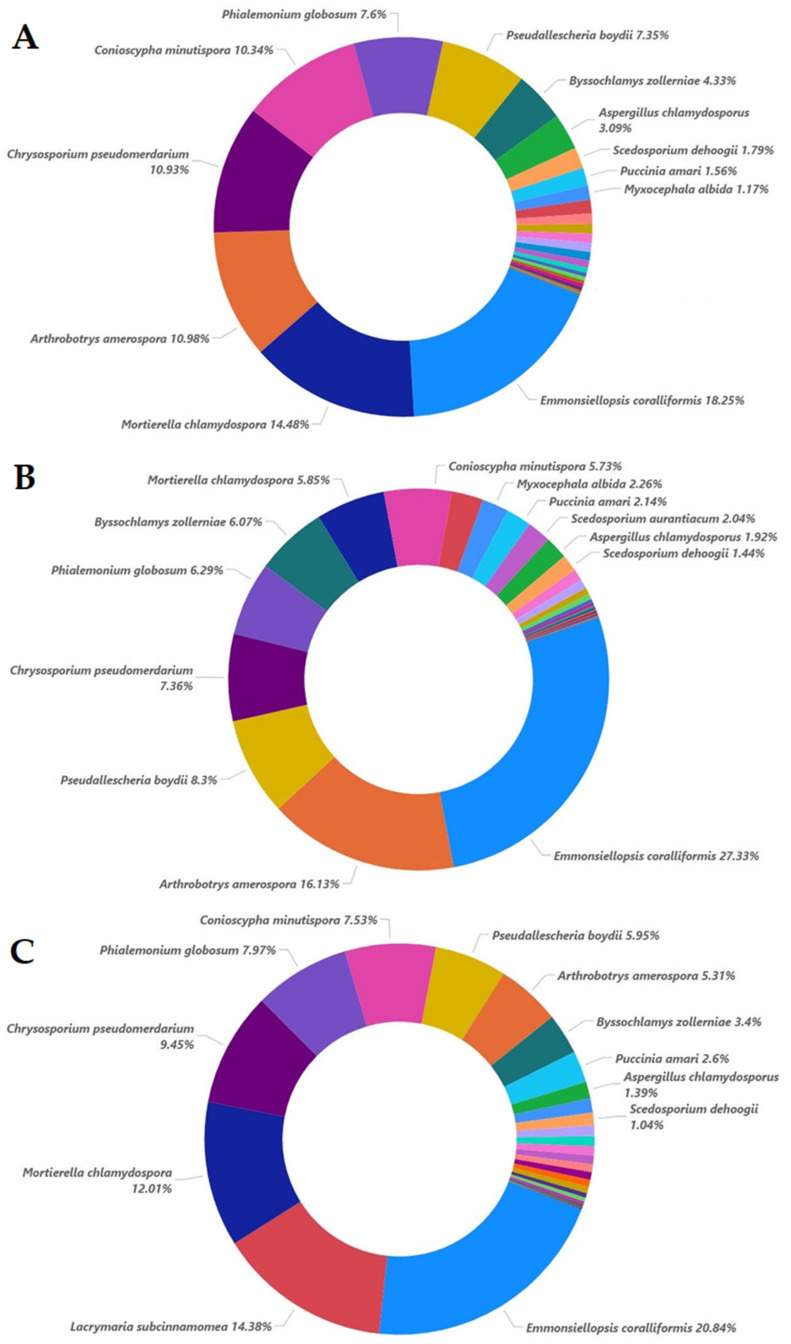
Relative species abundance with percentages higher than 1% for the non-inoculated samples at 0% (**A**), 5% (**B**), and 20% (**C**) essential oil concentrations.

**Figure 10 plants-12-00062-f010:**
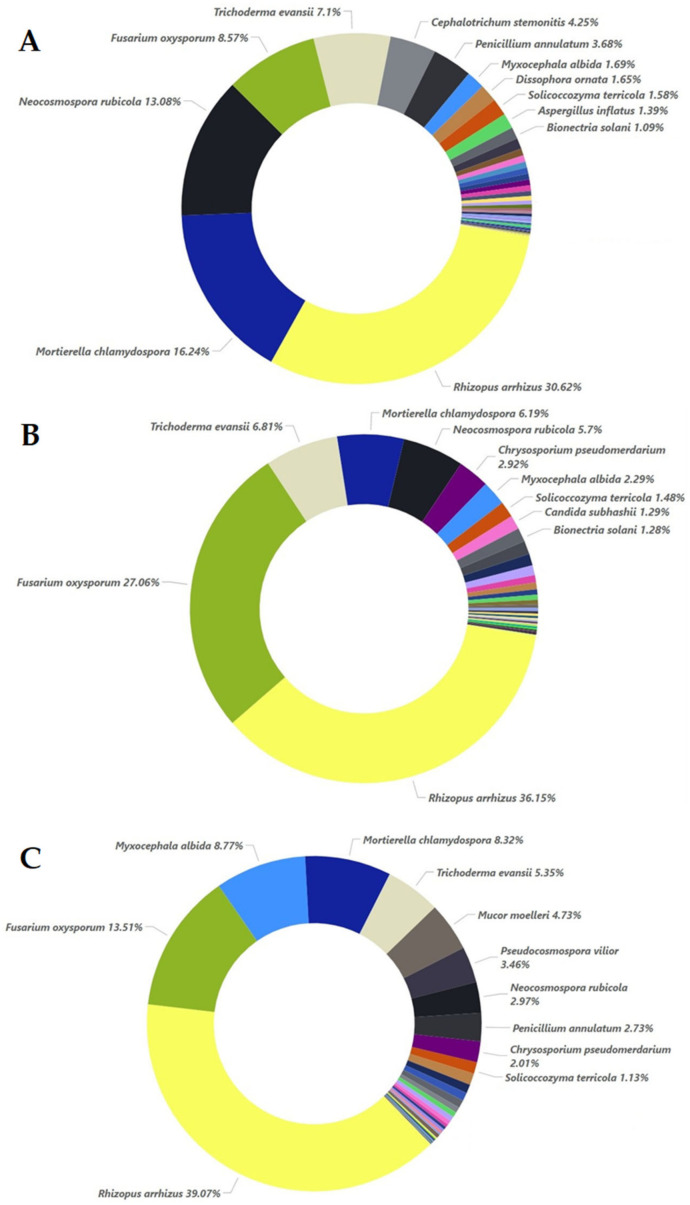
Relative species abundance with percentages higher than 1% for the inoculated samples at 0% (**A**), 5% (**B**), and 20% (**C**) essential oil concentrations.

**Figure 11 plants-12-00062-f011:**
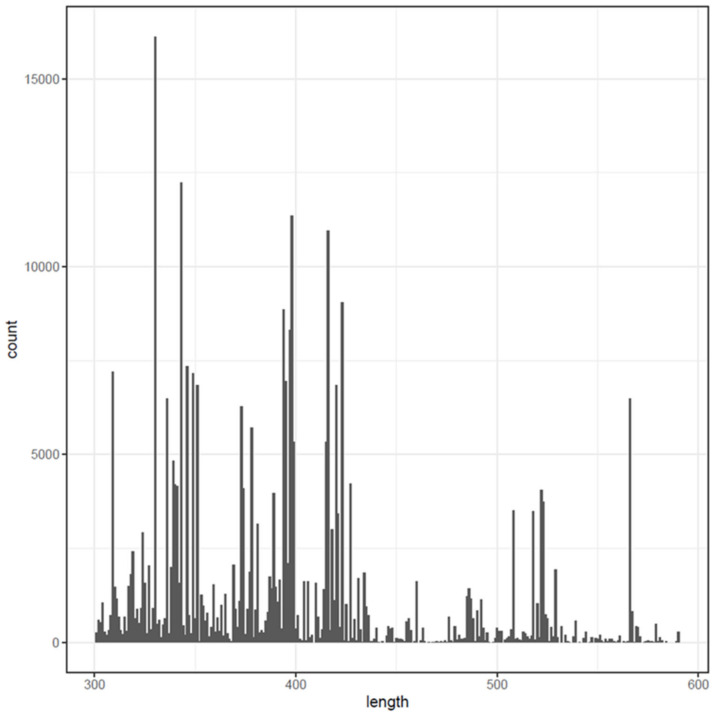
Sequence length distribution of the ITS1 and ITS2 regions for all the soil samples.

**Table 1 plants-12-00062-t001:** Reads per sample before and after the process of trimming and filtering.

Samples	Reads before Filtering	Reads after Filtering
A0	150,275	149,635
A5	122,035	121,498
A20	167,589	166,801
M0	137,604	136,998
M5	85,774	85,412
M20	103,173	102,805
Average	127,742	127,192
Total	766,450	763,149

**Table 2 plants-12-00062-t002:** Sequences of the primer sets (containing the Illumina overhang adapters) that were used for the amplification of the regions ITS1 and ITS2.

Region	Primer	Sequence
**ITS1**	Forward_1	5′- TCGTCGGCAGCGTCAGATGTGTATAAGAGACAGAGAGTTCATGCCCGAAAGGG-3′
	Reverse_1	5′-GTCTCGTGGGCT CGGAGATGTGTATAAGAGACAGCTGCGTTCTTCATCGAT-3′
	Forward_2	5′-TCGTCGGCAGCGTCAGATGTGTATAAGAGACAGGAAGGTGAAGTCGTAACAAGG-3′
	Reverse_2	5′-GTCTCGTGGGCTCGGAGATGTGTATAAGAGACAGAGCGTTCTTCATCGATGTGC-3′
**ITS2**	Forward_1	5′-TCGTCGGCAGCGTCAGATGTGTATAAGAGACAGATCGATG AAGAACGCAG-3′
	Reverse_1	5′-GTCTCGTGGGCTCGGAGATGTGTATAAGAGACAGGGTTTTGGCAGAAGCACACC-3′
	Forward_2	5′-TCGTCGGCAGCGTCAGATGTGTATAAGAGACAGCGTGAAGTGTCTTGCTGGTC-3′
	Reverse_2	5′-GTCTCGTGGGCTCGGAGATGTGTATAAGAGACAGCACCATACTTCGCGCAACAC-3′

## Data Availability

The data presented in this study are available on request from the corresponding author. The data are not publicly available due to privacy.

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
