# Peer review of "ITS Metabarcoding Reveals the Effects of Oregano Essential Oil on Fusarium oxysporum and Other Fungal Species in Soil Samples"

_plants, 2022, doi:10.3390/plants12010062_

Round 1

Reviewer 1 Report

1) Check English spelling.

2) There are too many graphs inside (there are 14 in total). A section of results with this number of graphs is too heavy to read.

3) They have not paid attention to putting the numbers of the figures and tables in order and within the manuscript they are not quoted correctly.

4) Discussion is too short and hasty not properly explaining why these results are obtained by giving an explanation.

5) The way it's written doesn't tell me anything.

L28: Change that with „those who”.

L29: Change were with “has been”.

L30: Change “was found to decrease” with “has decreased”.

L101: Change “Table 2” with “Table1”.

L112: I suggest to add inside the table the total and the average of the reads.

L116, L118, L119, L121, L132, L144, L147, L158, L161, L174, L177, L189, L190, L191, L194, L195, L215, L217-L225, L232-L234, L242-258, L266-L279, L306-L310, L312, L315, L317, L319, L322, L323, L326, L367, L414, L415: When you write treatment A change with non-inoculated treatment and for treatment M change.

L128, L142, L155: Change “per treatment (A up – M down)” with “for non-inoculated treatment (A) and soil-inoculated samples (B)”.  

L161: Change “Figure 6” with “Figure 4A”.

L166: Change “Figure 6” with “Figure 4B”.

L169: In this figure add both the figure from the treatment A and M. Add this description “Relative abundance with  percentages from Families taxa for non-inoculated treatment (A) and soil-inoculated samples (B)”.

L176: Change “Figure 7” with “Figure 5A”.

L180: Change “Figure 8” with “Figure 5B”.

L184: In this figure add both the figure from the treatment A5 and M5. Add this description “Relative abundance with  percentages from Families taxa at 5% concentration of essential oil for non-inoculated treatment (A) and soil-inoculated samples (B)”.

L194: Change “Figure 9” with “Figure 6A”.

L199: Change “Figure 10” with “Figure 6B”.

L203: In this figure add both the figure from the treatment A20 and M20. Add this description “Relative abundance with percentages from Families taxa at 20% concentration of essential oil for non-inoculated treatment (A) and soil-inoculated samples (B)”.

L214: Delete indices after diversity. Delete Figure 11.

L216:Change completely “More specifically”.

L223: What mean “invSimpson”?

L225: At the end of the sentence add  “(Figure 7)”.

L227: Change “Figure 10” with “Figure 7”.

L232: Change “Figure 12” with “Figure 8”.

L237: Change “Figure 11” with “Figure 8”.

L242: Change “Figure 13” with “Figure 9”.

L263: Change the description of graph. “Figure 9. Relative species abundance with percentages higher than 1% for the no-inoculated samples at 0% (A), 5% (B), and 20% (C) of essential oil.

L284: Change the description of graph. “Figure 10. Relative species abundance with percentages higher than 1% for the inoculated samples at 0% (A), 5% (B), and 20% (C) of essential oil.

L288: Strive????

L290: “Due to their simple and clear use”, this is not only the motivation…..

L299: Double space between oil and was.

L349: Change the Figure. Add only the figure 10 (for you are the figure 13). For me the Figure 14 not have sense in this contest.

L358: Double space between zollerniae and the.

L374: Double space between of and oregano.

L392-L397: This is results and you need to describe inside the results section.

L398: In this section you do not calculate the WHC and this is important for the laboratory experiment to maintain constant soil moisture. This aspect it is important for the microorganisms.

L409, L410: Delete treatment M and treatment A.

L423: Change table 1 with table 2.

L483: Change that with which.

Author Response

We would like to thank the reviewer for the comprehensive review of the manuscript and the suggestions which greatly improved the article.

1) Check English spelling.

Answer:  The text was edited by 2 native English-speaking people.

2) There are too many graphs inside (there are 14 in total). A section of results with this number of graphs is too heavy to read.

Answer: Indeed, there are many graphs, but we believe that the figures present the results in a more explanatory way, and it is easier for the reader to understand the relative changes in the microbial communities in inoculated and non-inoculated samples.

3) They have not paid attention to putting the numbers of the figures and tables in order and within the manuscript they are not quoted correctly.

Answer: We have corrected the numbers of the figures and tables and quoted them correctly in the manuscript

4) Discussion is too short and hasty not properly explaining why these results are obtained by giving an explanation.

Answer:  We have reviewed and edited the discussion section to better explain the obtained results

5) The way it's written doesn't tell me anything.

Answer:  We have reviewed and edited the discussion section to better explain the obtained results.

L28: Change that with „those who”.

Answer:  Done

L29: Change were with “has been”.

Answer: Done

L30: Change “was found to decrease” with “has decreased”.

Answer: Done

L101: Change “Table 2” with “Table1”.

Answer: Done

L112: I suggest adding inside the table the total and the average of the reads.

Answer: Done

L116, L118, L119, L121, L132, L144, L147, L158, L161, L174, L177, L189, L190, L191, L194, L195, L215, L217-L225, L232-L234, L242-258, L266-L279, L306-L310, L312, L315, L317, L319, L322, L323, L326, L367, L414, L415: When you write treatment A change with non-inoculated treatment and for treatment M change.

Answer: Done

L128, L142, L155: Change “per treatment (A up – M down)” with “for non-inoculated treatment (A) and soil-inoculated samples (B)”. 

Answer: Done

L161: Change “Figure 6” with “Figure 4A”.

Answer: Done

L166: Change “Figure 6” with “Figure 4B”.

Answer: Done

L169: In this figure add both the figure from the treatment A and M. Add this description “Relative abundance with percentages from Families taxa for non-inoculated treatment (A) and soil-inoculated samples (B)”.

Answer: Done

L176: Change “Figure 7” with “Figure 5A”.

Answer: Done

L180: Change “Figure 8” with “Figure 5B”.

Answer: Done

L184: In this figure add both the figure from the treatment A5 and M5. Add this description “Relative abundance with percentages from Families taxa at 5% concentration of essential oil for non-inoculated treatment (A) and soil-inoculated samples (B)”.

Answer: Done

L194: Change “Figure 9” with “Figure 6A”.

Answer: Done

L199: Change “Figure 10” with “Figure 6B”.

Answer: Done

L203: In this figure add both the figure from the treatment A20 and M20. Add this description “Relative abundance with percentages from Families taxa at 20% concentration of essential oil for non-inoculated treatment (A) and soil-inoculated samples (B)”.

Answer: Done

L214: Delete indices after diversity. Delete Figure 11.

Answer: Done

L216: Change completely “More specifically”.

Answer: Done

L223: What mean “invSimpson”?

Answer: “invSimpson” stands for ‘inverse Simpson”

L225: At the end of the sentence add  “(Figure 7)”.

Answer: Done

L227: Change “Figure 10” with “Figure 7”.

Answer: Done

L232: Change “Figure 12” with “Figure 8”.

Answer: Done

L237: Change “Figure 11” with “Figure 8”.

Answer: Done

L242: Change “Figure 13” with “Figure 9”.

Answer: Done

L263: Change the description of graph. “Figure 9. Relative species abundance with percentages higher than 1% for the no-inoculated samples at 0% (A), 5% (B), and 20% (C) of essential oil.

Answer: Done

L284: Change the description of graph. “Figure 10. Relative species abundance with percentages higher than 1% for the inoculated samples at 0% (A), 5% (B), and 20% (C) of essential oil.

Answer: Done

L288: Strive

Answer: Done

L290: “Due to their simple and clear use”, this is not only the motivation…..

Answer: The text “use along with their effectiveness against pathogens” was added.

L299: Double space between oil and was.

Answer: Done

L349: Change the Figure. Add only the figure 10 (for you are the figure 13). For me the Figure 14 not have sense in this contest.

Answer: Done

L358: Double space between zollerniae and the.

Answer: Done

L374: Double space between of and oregano.

Answer: Done

L392-L397: This is results and you need to describe inside the results section.

Answer: The text “The main constituents observed in the Origanum vulgare essential oil, by gas chromatography (GC) were: Carvacrol (51.5 %), p-Cymene (19.9 %), γ-Terpinene (8 %), Thymol (3.29 %), β-Caryophyllene (2.45 %), β-Myrcene (2.24 %), α-Terpinene (1.5 %), Sabinene (1.24 %), α-Pinene (1.2 %), β-bisabolene (1.2 %), d-limonene (0.97 %), borneol (0.84 %), Carvacrol methyl ether (0.65 %), α-Terpineol (0.58 %), β-Pinene (0.35 %), and α-Caryophyllene (0.29 %).” Has been moved to the results section “2.1 GC Analysis”

L398: In this section you do not calculate the WHC and this is important for the laboratory experiment to maintain constant soil moisture. This aspect it is important for the microorganisms.

Answer: The text “The soil water content was kept constant, by weighing the soil Petri dishes every two days and the amount of water that was lost, was added using spray bottles inside a laminar flow hood.” was added

L409, L410: Delete treatment M and treatment A.

Answer: Done

L423: Change table 1 with table 2.

Answer: Done

L483: Change that with which.

Answer: Done

Reviewer 2 Report

The authors presented a thorough investigation into the effect of oregano essential oil on F. oxisporum in soil. The fungal community in the soil was tested using a matabarcoding approach. For sequencing, they used two sets of primers for each of the ITS regions (ITS1 and ITS2). The results are very comprehensive and are graphically presented. The result showed that comparing a larger dose of oregano essential oil to a smaller amount, F. oxysporum growth was inhibited. Additionally, F. oxysporum dramatically changed the composition of other fungal species in soil.

However, for the use of oregano essential oil as a fungicide, additional and extensive research should be conducted.

I suggest publishing the manuscript because it covers a crucial area of inquiry into the usage of natural fungicides.

For correction:

I detected three extra spaces in Lines 292, 299, and 374.

Author Response

Answer

We would like to thank the reviewer for the valuable comments on our manuscript. Indeed we agree on the research needed and we are actually in the process of developing it.

We made all the requested changes suggested throughout the manuscript.

Round 2

Reviewer 1 Report

They answer all my request.

The English was improve.